# A Deep Neural Networks-Based Sound Speed Reconstruction with Enhanced Generalization by Training on a Natural Image Dataset

**Yoshiki Watanabe [1],*, Takashi Azuma [2] and Shu Takagi [3]**

1   Department of Bioengineering, School of Engineering, The University of Tokyo, Tokyo 113-8656, Japan
2   Lily MedTech Inc., Tokyo 113-8485, Japan; azumat@lilymedtech.com
3   Department of Mechanical Engineering, School of Engineering, The University of Tokyo, Tokyo 113-8656, Japan; takagi@mech.t.u-tokyo.ac.jp
*   Correspondence: y.watanabe@fel.t.u-tokyo.ac.jp

**Abstract:** Sound speed reconstruction has been investigated for quantitative evaluation of tissue properties in breast examination. Full waveform inversion (FWI), a mainstream method for conventional sound speed reconstruction, is an iterative method that includes numerical simulation of wave propagation, resulting in high computational cost. In contrast, high-speed reconstruction of sound speed using a deep neural network (DNN) has been proposed in recent years. Although the generalization performance is highly dependent on the training data, how to generate data for sufficient generalization performance is still unclear. In this study, the quality and generalization performance of DNN-based sound speed reconstruction with a ring array transducer were evaluated on a natural image-derived dataset and a breast phantom dataset. The DNN trained on breast phantom data (BP-DNN) could not reconstruct the structures on natural image data with diverse structures. On the other hand, the DNN trained on natural image data (NI-DNN) successfully reconstructed the structures on both natural image and breast phantom test data. Furthermore, the NI-DNN successfully reconstructed tumour structures in the breast, while the BP-DNN overlooked them. From these results, it was demonstrated that natural image data enables DNNs to learn sound speed reconstruction with high generalization performance and high resolution.

**Keywords:** breast cancer; deep learning; natural image; numerical simulation; ultrasound computed tomography

## 1. Introduction

Ultrasound imaging systems transmit ultrasound into the body and reconstruct cross-sectional images based on the information received from the scattering wave. They are commonly used in clinical situations because they are applicable in real-time, are cost-effective compared to X-ray CT and MRI, and do not require a dedicated room. B-mode imaging is the most popular and widely used ultrasound-based imaging technique. This method uses the nature of wave interference to reconstruct a map of scattering intensity. The intensity values of B-mode images are relative values based on a spatial gradient of the acoustic impedance of the tissue. Therefore, B-mode is a qualitative imaging method, limiting its diagnostic potential [1–4]. If implemented, quantitative imaging will help assess the efficacy of anticancer drugs on tumours over time and reduce operator-dependent inaccuracy in area and volume measurements [5–7].

In quantitative ultrasound imaging for the examination of breast cancer, sound speed imaging techniques using a ring array transducer have been investigated [8–20]. Greenleaf et al. [8] demonstrated the potential for detecting tumours in the breast by sound speed reconstruction based on transmitted waves, taking advantage of the fact that the tumour site is stiffer and has higher sound speed than other soft tissues. Subsequently, methods using

ray approximation [9,10], diffraction tomography that considers refraction and diffraction [11–13], and full waveform inversion (FWI) that directly considers the wave equation have been proposed for higher-quality (high-resolution and low-artifact) images [14–20]. Currently, methods based on FWI are mainstream as high-quality reconstruction methods based on physics.

On the other hand, FWI faces several challenges in its clinical application. One is the issue of computational cost: FWI is a method that optimises the sound speed distribution to create an observed signal in numerical simulations close to the real signal [14]. This optimisation process includes iterative calculations of the gradient method. To compute a gradient once, it is necessary to solve the forward problem of sound wave scattering twice. Therefore, the computational cost for convergence is significant and reconstruction cannot be carried out in real-time [16]. In addition, the problem is an inverse problem for a non-linear operator and has many local solutions [21]. Therefore, careful tuning is required in the design of the initial solution and regularisation parameters for implementation. These problems limit the clinical application of FWI.

Meanwhile, deep neural networks (DNNs) have been investigated for medical imaging application in recent years [22,23]. DNNs operate at high speed after training. Thus, the high-speed reconstruction of sound speed distribution using DNNs has been investigated. Several applications of DNNs in sound speed reconstruction have also been studied and it has been reported that sound speed distributions can be reconstructed from observed signals [24–26]. Fan et al. [24] and Prasad et al. [25] reported that reconstruction with DNNs trained on datasets generated by numerical simulation can be of higher quality than those with FWI, and that the DNNs are robust against noise. In addition, Feigin et al. [26] showed that reconstruction is possible even by signals obtained from single linear array transducer. These studies have shown promise for practical applications of sound speed reconstruction using DNNs.

The apparent performance of DNNs can vary significantly depending on how the training data and test data are chosen [27]. Therefore, DNNs require additional considerations compared to model-driven methods such as FWI. It can be expected that DNNs will show high performance when the training data contain the patterns of the test data. Conversely, there is a concern that performance may be significantly degraded when the training data do not contain the patterns of test data [28]. In previous studies, Prasad et al. [25] used numerical mediums composed of discs for training and test data. Feigin et al. [26] used numerical mediums composed of ellipses as training data and tested them on both numerical and physical phantoms of the same geometry. Fan et al. [24] used numerical breast phantoms for training data and tested on similar phantoms. Such evaluation systems may not accurately evaluate the generalization performance to out-of-distribution data.

From this viewpoint, Jush et al. [28] have examined how the performance of sound speed reconstruction by DNNs depends on the data domain. They showed that a DNN trained on breast phantom data performed poorly on elliptical data; similarly, a DNN trained on elliptical data performed poorly on breast phantom data. This result suggests that when DNNs are trained on datasets with limited patterns, such as breast phantom data and elliptical data, they do not perform well on data outside their domain. As a countermeasure to this low generalization performance, the mixing several types of datasets is proposed. However, this approach requires individual consideration for each application to determine the type and number of datasets to be prepared.

Typical breast datasets are produced by artificially assigning sound speeds to image data such as MRI images [20,24,28,29]. However, they do not necessarily correspond to the true sound speed of the tissue. In addition, since DNNs are trained to fit the domain of the training data, there is a risk wherein the output is predicted to have similar features to the artificial training data. Therefore, if estimation results are subjectively valid, there is a risk that they may not have physically meaningful sound speed. Another issue is data imbalance [30]. The main purpose of breast examination is to detect abnormal structures such as tumours, rather than normal tissue structures. However, data on such abnormal

structures are rare compared to normal structures, and it is difficult to collect enough data. Abnormal structures that appear less frequently are less important in training and are therefore expected to be inferior to the imaging ability of normal structures that appear more frequently [31].

For these reasons, there is a need to develop a single training dataset that contains sufficiently diverse patterns and achieves high generalization performance for unknown patterns, such as tumours. For this need, we propose a method to produce training data based on natural images [32–35]. There are many publicly available datasets of natural images, and their structures are far more diverse than the ones produced from breast phantoms or ellipse combinations. By utilizing these diverse structures in the training dataset's sound speed distribution, it may be possible to produce a single training dataset that leads to high generalization performance.

From the above, this study aims to develop a dataset based on natural images for a DNN that solves an inverse problem of sound speed reconstruction from observed signals. In addition, we aim to evaluate the generalization performance of the DNN trained on the natural image dataset in comparison to the case of the breast phantom dataset.

## 2. Materials and Methods

### 2.1. Generation of Sound Speed Distribution Datasets from Breast Phantom Data

Lou et al. [29] developed an open dataset of three-dimensional breast phantoms that can be used for optical and acoustic imaging. This dataset is based on MRI data and has an anatomically realistic breast structure with segmentation for each tissue. The dataset consists of data from three subjects with different breast densities. These data are three-dimensional and many 2D cross-sections can be extracted to produce training data. Each voxel is assigned an index indicating the tissue it belongs to. In this study, sound speed distributions were produced from the breast phantom data by the following procedures. First, sound speeds of 1515, 1478, 1615, and 1584 m/s were assigned for fibro-glandular tissue, fat, skin layer, and blood vessel, respectively [29,36]. The 1332 data sliced in the coronal plane were randomly split into training, validation, and test datasets in a ratio of 8:1:1. During training, the number of training data was increased to 80,000 by data augmentation through random horizontal flips and rotations at random angles. In addition, the breast phantom data were rescaled by bilinear interpolation to make the long sides of the breast data equal to the inner diameter of the ring array transducer.

### 2.2. Generation of Sound Speed Distribution Datasets from Natural Images

To produce training data in a medium rich in diverse patterns, sound velocity distributions were produced from a Google Open Images Dataset, which is one of the natural image datasets [32]. A total of 81,000 data pairs (training set: 80,000, validation set: 1000) were generated. The images were randomly rotated, cropped so that only the inner regions of the ring remained, and then divided into RGB channels. For each grid in the R channel, the brightness value (0–255) was assigned to a sound speed according to the following formula to generate a sound speed distribution.

$$c = c_{\min} + s\frac{(I - I_{\min})}{(I_{\max} - I_{\min})}(c_{\max} - c_{\min}), \tag{1}$$

where $c_{\min}$, $c_{\max}$ are the minimum and maximum values of the sound speed; $I$ is the brightness value; $I_{\min}$, $I_{\max}$ are the minimum and maximum of the brightness value; $s \in (0,1)$ is a parameter for scaling. The sound speed range $[c_{\min}, c_{\max}]$ was set to [1382, 1766] m/s, which is a doubling of the range of sound speed in biological soft tissues from 1478 m/s in fat, which has low sound speed, to 1670 m/s in ligaments, which has high sound speed [36]. $s$ was sampled from a uniform distribution in the interval (0,1) for each image individually and was introduced to make the generated sound speed distribution more diverse.

### 2.3. Computation of the Observed Signals

Estimation of the sound speed distribution from the observed signals is an inverse problem. Hence, it is necessary to obtain observed signals corresponding to the sound speed distribution produced in the previous section for the dataset. The observed signals are obtained by solving the forward problem of scattering of sound waves. This section describes how to discretise the Helmholtz equation and solve the forward problem. The scattering of sound waves in a sound speed inhomogeneous medium is represented by the Helmholtz equation as follows.

$$\left(\nabla^2 + \frac{\omega^2}{c^2}\right)p = -s, \tag{2}$$

where $c$ is the sound speed, $p$ is the pressure field, $s$ is the wave source, and $\omega$ is the angular frequency. Let $N_x$ and $N_y$ be the number of grids in the x and y directions in the two-dimensional space, the discretized Helmholtz equation can then be expressed by the following equation.

$$AP = -S, \tag{3}$$

where $A$ is the $(N_x N_y \times N_x N_y)$ matrix of discretized coefficients in parentheses on the left side of Equation (3) that considers the absorbing boundary conditions, $P$ is a discretized pressure field vector with $(N_x N_y \times 1)$ components, and $S$ is a $(N_x N_y \times 1)$ matrix representing a wave source. An example row of $A$ is shown in the following expressions: Formulas (4) and (5). Note that in this example, for simplicity, the second-order difference in the spatial direction is implemented with second-order precision.

Let $\ell = i + jN_x$, the formula for $\ell$-th row of $A$ on Equation (3) is expressed as follows.

$$\sum_{m=1}^{N_x N_y} A_{\ell,m} P_m = -S_\ell. \tag{4}$$

$A_{\ell,m}$ is expressed as follows.

$$A_{\ell,m} = \begin{cases} \frac{1}{\Delta x^2}, & \text{for } m = (i-1) + jN_x, \ (i+1) + jN_x, \ i + (j-1)N_x, \ i + (j+1)N_x \\ \left(\frac{-4}{\Delta x^2} + \frac{\omega^2}{c(i,j)^2}\right), & \text{for } m = i + jN_x \\ 0, & \text{otherwise} \end{cases}, \tag{5}$$

where $\ell = i + jN_x$ and $c(i,j)$ is the sound speed at the point $(i,j)$. In this study, the terms corresponding to the second-order partial derivatives in the spatial direction of $A$ were implemented using fourth-order central difference discretization, and the absorbing boundary conditions were implemented using perfectly matched layers. The pressure field $P$ was obtained by LU-decomposition of $A$ and inverting it for the system of equations.

### 2.4. Network Architecture and the Training

We implemented DNNs using TensorFlow 2.1. The DNNs had an encoder–decoder type architecture based on a convolutional neural network (CNN), where the input was the real and imaginary parts of the observed signal, and the output was the sound speed distribution (Figure 1). The backbone of the architecture is based on ResNet [37] and transfer learning was not conducted. The mean absolute error (MAE) of the sound speed distribution was used as the loss function. For optimization, we employed the Adam optimizer [38] with a batch size of 2 during training. The number of training epochs was 600, and the weights were chosen for minimum loss for the validation set.

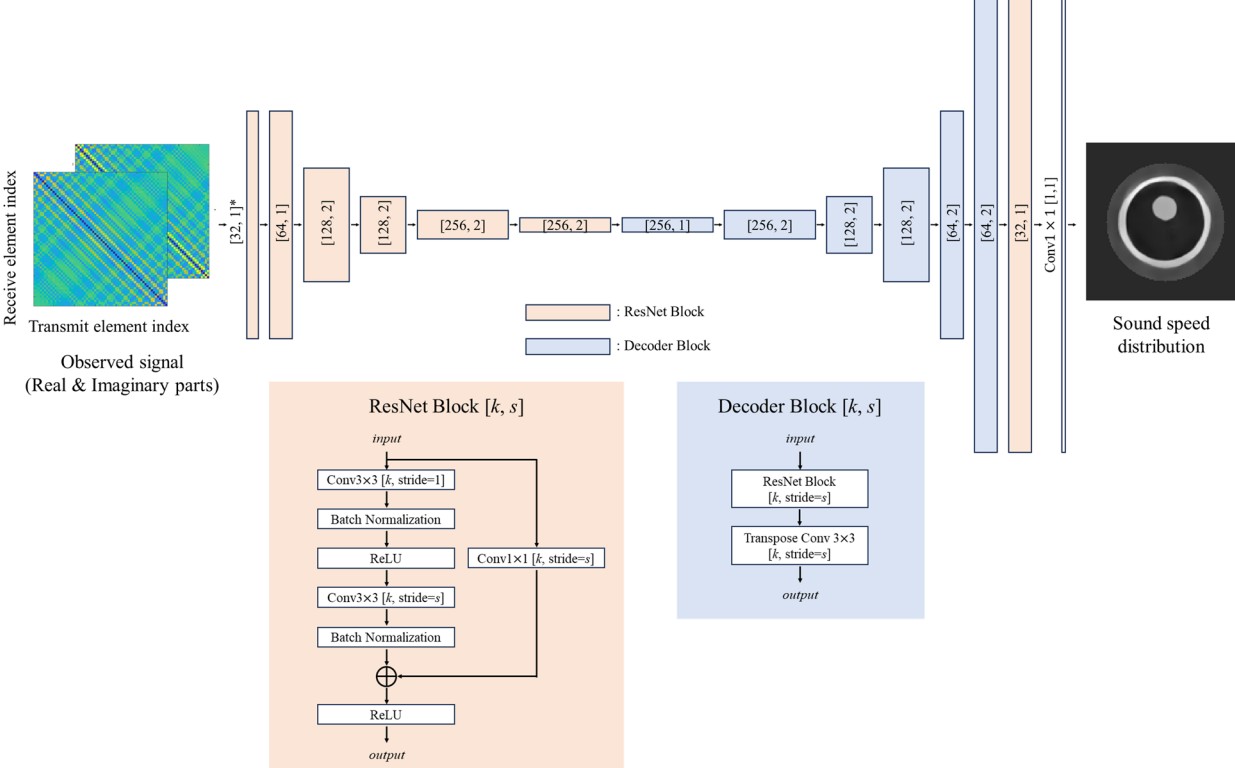

**Figure 1.** Architecture of the ResNet-based Network. Input: observed signal (real and imaginary parts for all receiver/transmitter pairs). Output: sound speed distribution. The numbers in square brackets denote the number of output filters in the convolution and the stride. Upper: overview of the ResNet-based encoder–decoder architecture, which consists of a sequence of ResNet blocks and decoder blocks. Kernel size of the first convolution layer of ResNet block with * is modified from $3 \times 3$ to $7 \times 7$. Lower left: ResNet block consists of two $3 \times 3$ 2D convolutional layers, with ReLU and batch normalization operations, and a residual connection with $1 \times 1$ convolutional layer. Lower right: decoder block consists of a ResNet block and a $3 \times 3$ transpose convolutional layer.

### 2.5. Measurement Condition

A ring array transducer, which is commonly used in sound speed reconstruction, was used. The advantage of a ring array transducer is that it can acquire transmission waves, whereas linear array transducers and other one-sided probes can only acquire tissue scattering waves. The conditions of the measurement system are shown in Table 1. Computational cost is an issue as tens of thousands of data are required for the training of the DNNs. Therefore, the physical size of the system was reduced in this study. Figure 2 shows the element arrangement.

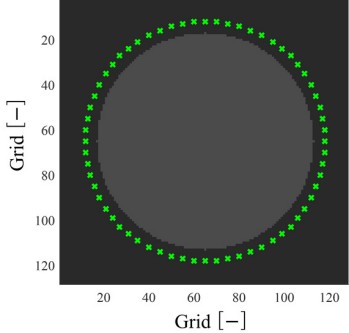

**Figure 2.** Array geometry. Green crosses indicate transducer positions. Gray region indicates the reconstruction area.

**Table 1.** Condition of the transducer.

| Ring diameter | 20 mm |
|---|---|
| Number of elements | 64 |
| Frequency | 500 kHz |
| Grid size | 187.5 μm |

## 3. Results

### 3.1. Generated Datasets and the Properties

Figure 3 shows examples of the data generated. The data based on natural images show spatially diverse structures in the sound speed distribution. On the other hand, the data based on the breast phantom show sound speed distributions which have similar patterns each other. Therefore, to compare the diversity of the structures, the datasets were evaluated from a spatial frequency perspective. The Fast Fourier Transform (FFT) was applied to the sound speed distributions of each dataset. A rotationally symmetric 2D Hanning window with a window width equal to the ring inner diameter was used. Furthermore, a one-dimensional frequency spectrum was obtained by averaging in frequency space in the circumferential direction. The sum of the amplitudes of the spatial frequency components at wavelengths where they are less than half of the ultrasound wavelength in this spectrum and those above half of the ultrasound wavelength were plotted on a two-dimensional plane (Figure 4). Here, constant components and spatial frequency components with wavelengths larger than the ring diameter were excluded from the analysis. The figure shows that the sound speed distribution based on natural images has a broad distribution that includes that of the breast phantom. This result shows that the sound speed distribution based on natural images has a diverse structure in terms of spatial frequencies.

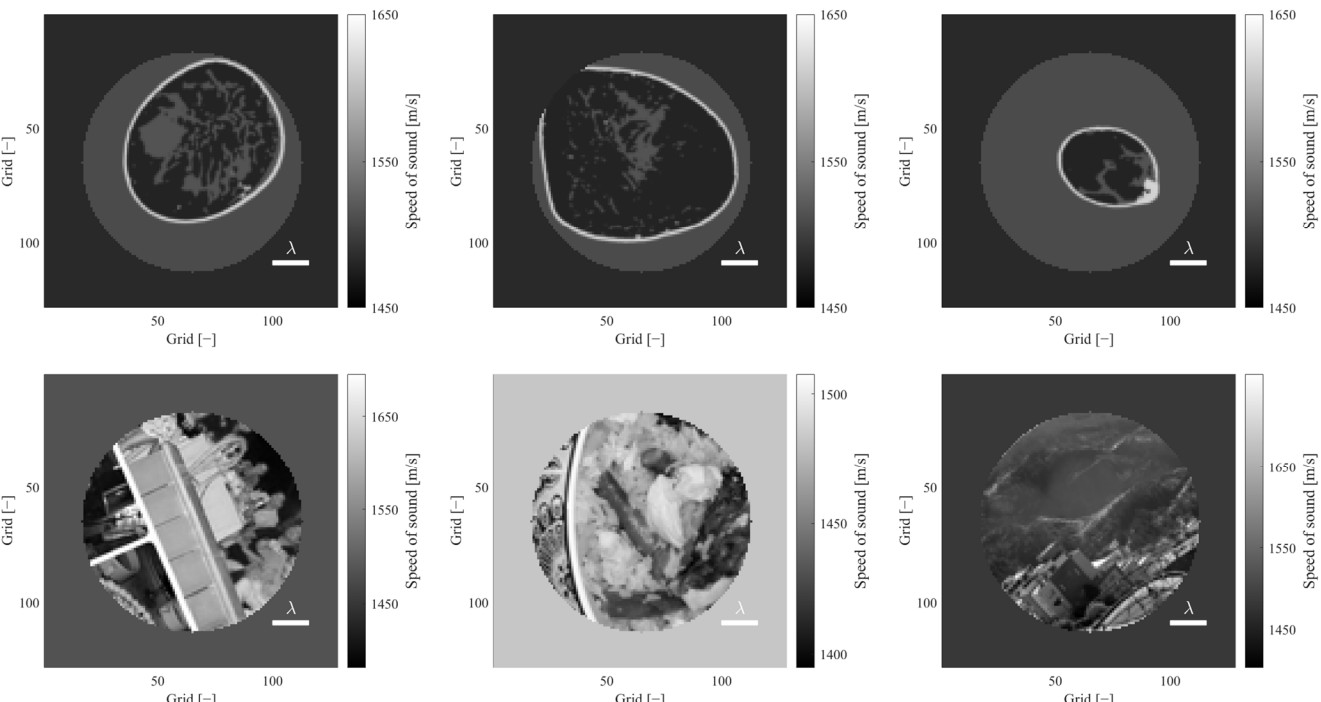

**Figure 3.** Example of generated training data. Upper row: generated from breast phantom dataset. Bottom row: generated from natural image dataset. Scale bar in images shows wavelength of the ultrasound.

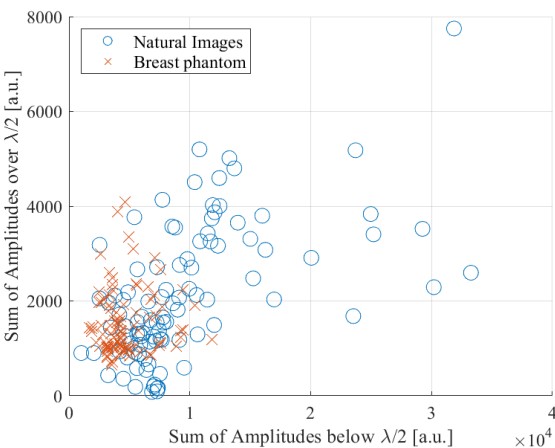

**Figure 4.** Comparison of spatial frequency distribution. Horizontal axis shows sum of spatial frequency spectrum for spatial wavelength below ultrasound wavelength. Vertical axis shows sum of spatial frequency spectrum for spatial wavelength over ultrasound wavelength. A total of 100 samples were randomly extracted from each dataset.

### 3.2. Visual Comparison

Figure 5 shows images reconstructed by the DNNs. First, the reconstruction quality of the breast phantom test data is described. The DNN trained on the breast phantom data (BP-DNN) reconstructed fibroglandular tissue and skin structures in detail. Although some structures much smaller than a wavelength are lacking, we still observe that the structures around 1/10 wavelength, such as the skin, were reconstructed. On the other hand, the DNN trained on natural image data (NI-DNN) seemed to reconstruct blurred images compared to the images reconstructed by the BP-DNN. However, the NI-DNN has sufficient resolution for practical use in the reconstruction of breast structures because the NIDNN had a resolution of about half a wavelength.

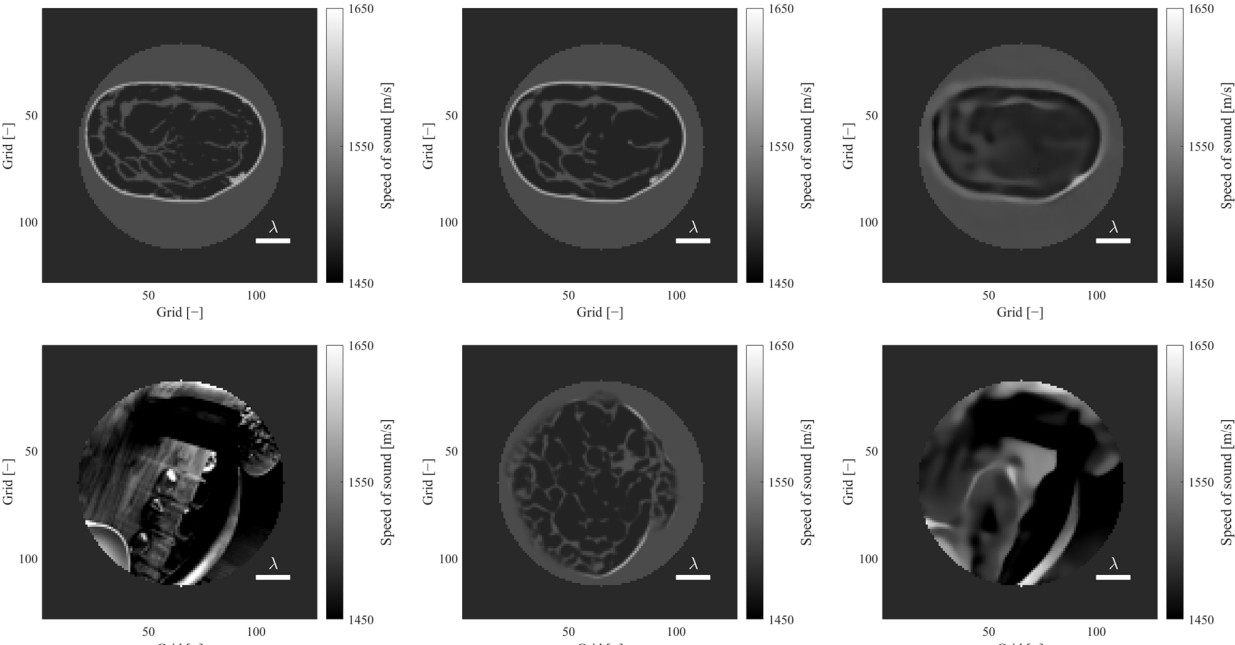

**Figure 5.** Examples of predicted sound speed distributions. Upper row: breast test data. Bottom row: natural image test data. Left column: ground truth. Centre column: prediction of a DNN trained on breast phantom dataset (BP-DNN). Right column: prediction of a DNN trained on natural image dataset (NI-DNN). BP-DNN prediction for a natural image test data shows false structure (Bottom middle).

In addition, the reconstruction quality of the natural image test data is described. The BP-DNN produced false structures that look like deformed breast structures resembling the ground truth. This may be due to the low generalization performance to out-of-domain structures, as the BP-DNN was trained only on the breast structure. On the other hand, the NI-DNN reconstructed about a half-wavelength structure. The NI-DNN is considered to have higher generalization performance than the BP-DNN, even for complex structures.

### 3.3. Quantitative Evaluation

For quantitative evaluation of the reconstruction performance, the mean absolute error (MAE) was calculated for each training/test dataset pair by the following formula (Table 2).

$$MAE = \frac{1}{N} \sum_{x_i,\, y_i \in B} \left| c_{\text{pred}}(x_i, y_i) - c_{\text{true}}(x_i, y_i) \right|, \tag{6}$$

where $N$ is the number of grids in the reconstructed region, $B$ is the set of grids in the reconstructed region, $c_{\text{pred}}$ is the predicted sound speed by a DNN, and $c_{\text{true}}$ is the true sound speed. The mean absolute error (MAE) was calculated for each sample. Subsequently, the mean and standard deviation of the MAE across the entire dataset were evaluated. The BP-DNN was able to predict the sound speed distribution with a small MAE for the breast phantom test data. However, the average MAE for the natural image data was about 50 times larger than that for the breast phantom test data. This suggests that the DNN, when trained on breast phantom data, struggles to generalize to more diverse patterns, such as those found in natural image data. On the other hand, the NI-DNN showed a larger error in the breast test data than the BP-DNN did. However, the MAE for the breast phantom data is 5.5 m/s, whereas typical tumours have a sound speed difference to the surrounding tissue of around 30–50 m/s. Therefore, this DNN is considered to have the required sound speed accuracy for tumour evaluation. Together with the subjective visual evaluation, the NI-DNN is considered to have a high generalization performance that is applicable to breast data as well.

**Table 2.** The mean absolute error comparison [m/s] (Mean $\pm$ SD).

|  | Breast (Train) | Natural Images (Train) |
| --- | --- | --- |
| Breast (Test) | 1.4 $\pm$ 3.8 | 5.5 $\pm$ 9.6 |
| Natural images (Test) | 72.5 $\pm$ 44.1 | 9.6 $\pm$ 15.3 |

### 3.4. Generalization Performance Evaluation for Tumour Structures outside the Training Data

DNNs are required to have the ability to image abnormal structures such as tumours in the breast. To assess the generalization performance against abnormal structures, we produced three different tumour models with different sizes (about 0.2, 0.8, 1.3 wavelength) at a sound speed of 1548 m/s on a breast phantom. A comparison of the reconstructed images is shown in Figure 6. The figure shows that the BP-DNN hardly reconstructs tumours that are not in the training data. For the largest tumour, the BP-DNN seems to attempt to mimic the tumour by concentrating the fibro-glandular tissue.

On the other hand, the NI-DNN accurately reconstructed tumours with sizes of 0.8 and 1.3 wavelengths, although reconstruction for the smallest tumours was blurred. Even though there are no such data in the natural image dataset of tumours in the breast, the NI-DNN successfully reconstructed the structures down to about half a wavelength. We attribute its success to the fact that the natural image dataset offered a broad pattern of structures, which allowed for the NI-DNN to acquire more physically reasonable reconstructions compared to the BP-DNN.

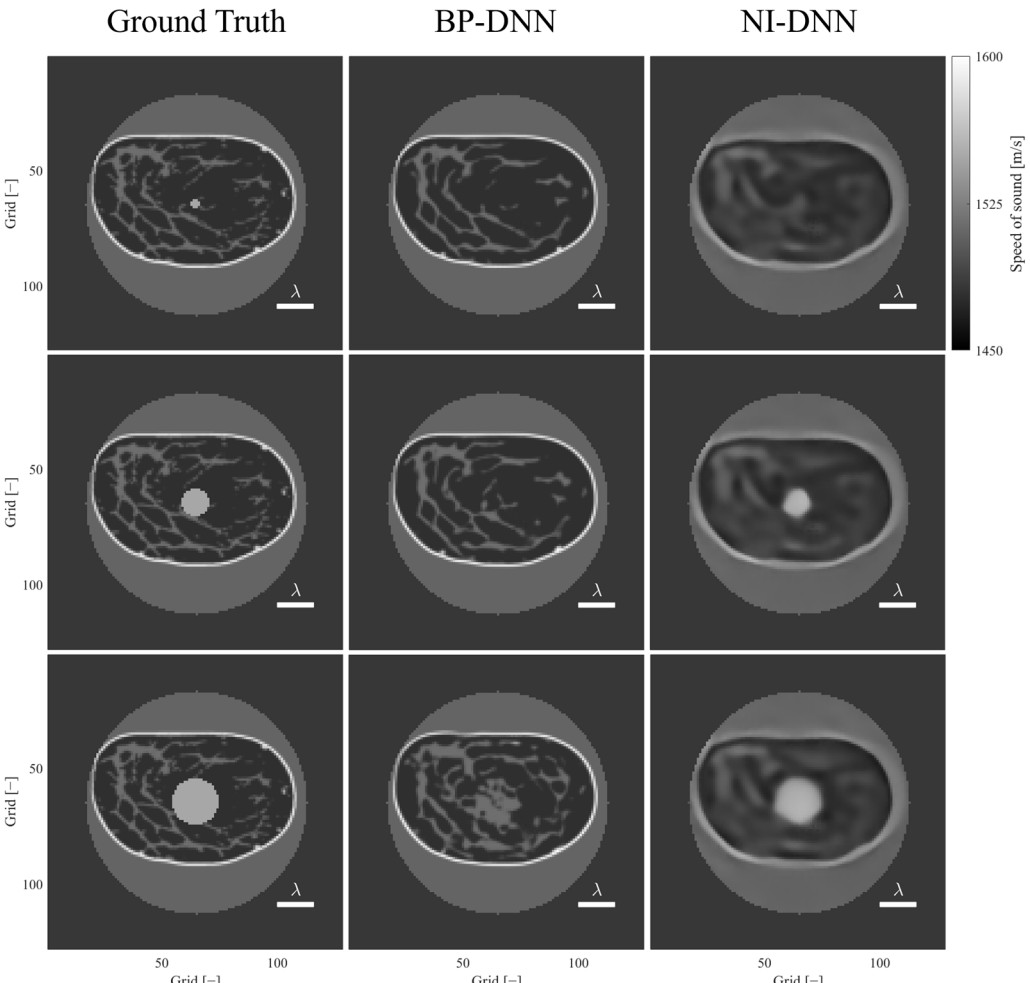

**Figure 6.** Generalization performance against tumours in the breast. Each row shows a ground truth and the DNNs predictions of the breast model, according to their tumour sizes. Left column: ground truth. Centre column: prediction of the BP-DNN. Right column: prediction of the NI-DNN. The NI-DNN could capture the tumours with a resolution of about half a wavelength, while the BP-DNN overlooked the tumours.

## 4. Discussion

### 4.1. Impacts of Training Data on Reconstruction Quality and Generalization Performance

We discuss the performance difference between the BP-DNN and NI-DNN. First, we discuss the performance of the BP-DNN. From the visual evaluation, the BP-DNN was able to reconstruct structures down to 1/10 of a wavelength. This is finer than 1/2 wavelength, which is the minimum resolution (diffraction limit) for imaging using the nature of wave interference. The reasons and limitations of this phenomenon are discussed from the perspective of the constraints imposed on the task.

Model-driven methods, such as FWI, utilise physical models to perform sound speed reconstruction. In this case, the constraints (prior information) of the problem are given in a formulated form, such as total variation regularisation [39] and Tikhonov regularisation [40]. On the other hand, DNNs are data-driven methods and utilise the constraints estimated from the training data to perform sound speed reconstruction. In this case, there is no need to provide additional constraints by regularisation. The BP-DNN in this study is considered to have estimated the constraints of the problem from the training data and restricted its solution space to a smaller subspace. Therefore, it is assumed that the BP-DNN is able to provide detailed texture from the measured signal even at a large wavelength relative to the structure.

This approach is effective when there is no domain shift, i.e., the training data and the real-world data are sampled from the same distribution [41]. However, as discussed in the introduction, the true sound speed distributions of breasts are unknown. Even if the true sound speed distribution could be obtained by other methods, it would be difficult to collect a large amount of data on abnormal structures such as tumours, which are important in medical examinations. This means that it is difficult to avoid a domain shift. Therefore, there is concern that the DNN may not work properly if there is a difference between the artificially produced breast phantom and the actual breast in pattern of sound speed distribution, or if there is a tumour that appears infrequently. Actually, our experimental results showed the BP-DNN had poor generalization performance, as it was unable to reconstruct structures and produced false structures for out-of-domain data (tumour models and natural image data). This suggests that the training on the breast phantom data led to an insufficient model in generalization performance.

On the other hand, the NI-DNN showed high generalization performance. The NI-DNN was able to reconstruct structures not only on the test data of natural image data, but also on the breast phantom and tumour models data which have patterns not included in the training data. Although its resolution was inferior to the BP-DNN, it was able to reconstruct structures down to half a wavelength, which is fine enough for practical use. These results can be explained as follows. The BP-DNN has an excessively restricted solution space, which means that more accurate solutions can be reached for the data inside the training data domain, resulting in very fine resolution. Simultaneously, the generalization performance is significantly degraded for out-of-domain data. On the other hand, the NI-DNN with a wide variety of patterns is assumed to have a large solution space, so the range of possible solutions is wide, resulting in a resolution that is not extremely small. Simultaneously, the generalization performance is improved for data with patterns not included in the training data.

Since it is important that abnormal out-of-domain structures can be reconstructed without being overlooked in sound speed imaging in breast examination, training data that can improve generalization performance are more suitable as the resolution of about half a wavelength can be kept. From the above discussion, natural image data are more suitable for training data for breast examination than artificial breast phantom data.

*4.2. Towards Reliable Sound Speed Imaging*

In FWI, a model-driven approach that has been studied in sound speed imaging, the sound speed distribution is optimised so that the error between the real observed signal and the numerically simulated one is minimised [14–20]. Thus, the reliability of the solution can be assessed from the error. On the other hand, such evaluation is not available in sound speed imaging with DNNs. In addition, as seen in the experimental results, DNNs can produce false structures that look like the real thing. Therefore, sound speed imaging with DNNs for the purpose of medical examination can be very risky. To reduce this risk, it is necessary to develop methods to improve the reliability of examination, in addition to the generalization performance considerations that have been discussed in this study. There are two possible ways to do this.

The first method is to add an out-of-distribution detection system, which assesses whether the input data to a DNN are outside the distribution of the training data [42,43]. For example, by comparing the distribution of training data and input data in the feature space of the middle layer of the DNN, the system can tell the operator whether the prediction is reliable or not.

The second method is to add a step to solve the forward problem, such as FWI. After prediction of the sound speed distribution with a DNN for solving the inverse problem, we acquire the observed signal for the predicted sound speed distribution. By comparing this observed signal with that of the original input, the validity of the predicted sound speed distribution can be assessed. The forward problem can be computed by numerical

simulation, but real-time performance will be an obstacle, so a new DNN for solving the forward problem should be prepared.

False structures, such as those seen in Figure 5, can be detected by the methods described above. Investigating these methods can improve the reliability of imaging.

### 4.3. Limitations

Limitations of this study are described. In the experiments, the array transducer had a smaller ring diameter than that of clinical use. Our previous study [44] has explained that the adoption of small ring diameters in DNNs has greater advantages in significantly reducing computational cost and enabling the generation of large amounts of training data, because DNNs do not suffer from cycle-skipping as seen in FWI. In addition, as wavelength mainly determines image quality, a small ring diameter does not have a significant impact on image quality. From the above, we consider that the use of the small ring diameter is reasonable in this study.

Second, we discuss the limitations regarding the consideration of training data and generalization performance. In Section 4.1, we noted that one of the reasons the BP-DNN failed to adapt to out-of-domain data was the limited patterns in the breast phantom dataset. There are several possible ways to improve pattern diversity in the breast phantom dataset. For example, providing variation in the sound speed assigned to each tissue may improve generalization performance for abnormal sound speed structures. However, in this case, the pattern of possible solutions increases, which may lead to decreased resolution. Another possible way to improve generalization performance is to add tumour models to the training data. In this case, the tumour shape and sound speed settings need to be investigated. As seen in this study, DNNs tend to create the output close to the pattern of the training data. For example, if a DNN is trained with circular tumours, it may reconstruct a tumour having a different shape as a circular one. In other words, introducing simple tumour models may result in imaging that does not reflect the true tumour characteristics. Since it is difficult to obtain patterns of true tumour sound speed distribution, additional investigation is required for these settings. These considerations should be taken into account for the production of a more appropriate dataset.

## 5. Conclusions

In this study, we developed a dataset based on natural images for a DNN that solves an inverse problem of sound speed reconstruction. Through visual and spatial frequency evaluations, we found that the developed dataset has more diverse patterns than those of the breast phantom dataset. Furthermore, we evaluated the generalization performance of DNN-based sound speed reconstruction for the developed dataset compared to a breast phantom dataset. The DNN trained on the natural image data (NI-DNN) successfully reconstructed both natural image and breast phantom test data with a resolution of about half a wavelength, while the DNN trained on the breast phantom data (BP-DNN) could not reconstruct natural image test data. In addition, the BP-DNN exhibited issues, as it could produce false structures and was unable to reconstruct tumour structures. In contrast, the NI-DNN did not produce any apparent false structures and was able to reconstruct tumours on the breast phantom. These results indicate that the developed natural image-derived dataset enable DNNs to learn sound speed reconstruction with high generalization performance and high resolution.

**Author Contributions:** Conceptualization, Y.W. and T.A.; Methodology, Y.W.; Software, Y.W.; Validation, Y.W. and S.T.; Formal analysis, Y.W.; Investigation, Y.W.; Resources, Y.W. and S.T.; Data curation, Y.W.; Writing—original draft, Y.W.; Writing—review & editing, T.A. and S.T.; Visualization, Y.W.; Supervision, S.T.; Project administration, S.T. All authors have read and agreed to the published version of the manuscript.

**Funding:** This research received no external funding.

**Institutional Review Board Statement:** Not applicable.

**Informed Consent Statement:** Not applicable.

**Data Availability Statement:** The breast models used in this study were sampled from the OA-Breast Database created by the Computational Imaging Science Laboratory at Washington University in St. Louis [29]. This data was made available under the Open Database License and can be found at https://anastasio.bioengineering.illinois.edu/downloadable-content/oa-breast-database/ (accessed on 12 November 2023). The natural images dataset used in this study were sampled from the Open Images Dataset created by Google [32]. This data was made available under the CC BY 2.0 license and can be found at https://storage.googleapis.com/openimages/web/index.html (accessed on 12 November 2023).

**Acknowledgments:** We thank Hongxiang Lin from Research Center for Healthcare Data Science, Zhejiang Lab, Hangzhou, China for knowledge of full waveform inversion.

**Conflicts of Interest:** Author Takashi Azuma was employed by the company Lily MedTech Inc. The remaining authors declare that the research was conducted in the absence of any commercial or financial relationships that could be construed as a potential conflict of interest.

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
