# Peer review of "A Deep Neural Networks-Based Sound Speed Reconstruction with Enhanced Generalization by Training on a Natural Image Dataset"

_applsci, doi:10.3390/app14010037_

Round 1

Reviewer 1 Report

Comments and Suggestions for Authors

Dear Editor,

I would like to thank you the opportunity to review the manuscript entitled “A deep neural networks-based sound speed reconstruction with enhanced generalization by training on a natural image dataset” and so contribute with this Journal.

The present manuscript aimed to develop a dataset based on natural images for a deep neural network (DNN) that solves an inverse problem of sound speed reconstruction from observed signals and to evaluate the generalization performance of the DNN trained on the natural image dataset in comparison to the case of the breast phantom dataset. Although there are more than 1900 articles published on the topic, this manuscript offers a valuable contribution. Its structure is impeccable. I believe that the topic presented is of interest to the Journal's readers. The methodology is correct and allows the authors to answer the questions asked. The manuscript needs small changes so that it can be indicated for publication.

See below my observations about the manuscript.

Introduction

- First paragraph (page 1): I suggest a new format. The paragraph is long several statements are made. However, no references are presented to support these statements. Therefore, I suggest including references.

- Second paragraph (page 2, lines 50-59): Please insert references that support the statements made in this paragraph.

- Third paragraph (page 2, lines 71-76): Please insert references that support the statements made in this paragraph.

- Fifth paragraph (page 2, lines 92-102): Please insert references that support the statements made in this paragraph.

- First paragraph (page 3, lines 103-111): Please insert references that support the statements made in this paragraph.

- Fist paragraph (page 3, line 108): “We develop a method to convert...”. Did the development of the method result in publication? If yes, cite the study.

 Discussion

- The observations made in the introduction are valid for discussion. The absence of citations casts doubt on the statements made.

Conclusions

- Greater objectivity could have been given in the conclusion. I suggest a new wording. This new format should be more direct, seeking to respond to the objectives presented.

 References

- I understand the importance of citing classic works. However, would it be possible to present more current references?

Author Response

Comment 1: (Introduction) First paragraph (page 1): I suggest a new format. The paragraph is long several statements are made. However, no references are presented to support these statements. Therefore, I suggest including references.

Response 1: Thank you for pointing out the lack of references in the first paragraph. We have now added appropriate citations to support the statements made. These references include (Spratt et al., 1996, Vriens et al., 2016, Kim et al., 2021.) to provide a strong foundation for our claims. (page 1, line 40)

Comment 2: (Introduction) Second paragraph (page 2, lines 50-59): Please insert references that support the statements made in this paragraph.

Response 2: As suggested, we have inserted references in the second paragraph to support our statements. The added references are (Sandhu et al., 2015, Agudo et al., 2017, Robins et al., 2021.), which align with the context and content of our discussion. (Page 2, lines 51-60, highlighted.)

Comment 3: (Introduction) Third paragraph (page 2, lines 71-76): Please insert references that support the statements made in this paragraph.

Response 3: As suggested, we have inserted references in the third paragraph to support our statements. The added references are (Farahani et al., 2020, Jush et al, 2023.), which align with the context and content of our discussion. (Page 2, lines 73, 77, highlighted.)

Comment 4: (Introduction) Fifth paragraph (page 2, lines 92-102): Please insert references that support the statements made in this paragraph.

Response 4: As suggested, we have inserted references in the Fifth paragraph to support our statements. The added references are (Lin et al., 2016, Fan et al., 2022, Jush et al., 2023, Lou et al., 2017., Johnson et al., 2019, Bria et al., 2020.), which align with the context and content of our discussion. And the term 'data unbalance' has been changed to 'data imbalance' for consistency with the cited literature. (Page 2, lines 93–104, highlighted.)

Comment 5: (Introduction) First paragraph (page 3, lines 103-111): Please insert references that support the statements made in this paragraph.

Response 5: As suggested, we have inserted references in the first paragraph to support our statements. The added references are (Kuznetsova et al., 2020, Deng et al., 2009, Krizhevsky et al., 2009, Lin et al., 2014.), which align with the context and content of our discussion. (Page 3, line 108, highlighted.)

Comment 6: (Introduction) Fist paragraph (page 3, line 108): “We develop a method to convert...”. Did the development of the method result in publication? If yes, cite the study.

Response 6: Thank you for pointing this out. The development is the first time it is done in this paper, so we have rewritten it to make it clear. (Page 3, line 110-112, highlighted.)

Comment 7: (Discussion) The observations made in the introduction are valid for discussion. The absence of citations casts doubt on the statements made.

Response 7: Thank you for your advice. As your suggestion, we have inserted references in the discussion. The added references are (Rudin et al., 1992, Tikhonov et al., 1977, Hendrycks et al., 2016, Lee et al., 2018, Hsu et al., 2020) and references for FWI (Sandhu et al., 2015, Roy et al., 2010, Agudo et al., 2017, Li et al., 2014, Wang et al., 2015, Pérez et al., 2017, Lin et al., 2016.). (Page 10, lines 282, 283, 291, 323, 332, highlighted.)

Comment 8: (Conclusions) Greater objectivity could have been given in the conclusion. I suggest a new wording. This new format should be more direct, seeking to respond to the objectives presented.

Response 8: Thank you for your suggestions. We have made the conclusion more objective and clarified the correspondence with the objectives of the study that stated in the Introduction. (Page 11, line 371-384.)

Comment 9: (References) I understand the importance of citing classic works. However, would it be possible to present more current references?

Response 9: Thank you for your advice. I added many more current references that include (Kim et al., 2021, Robins et al., 2021, Farahani et al., 2021, Johnson et al., 2019, Bria et al., 2020, Hendrycks et al., 2016, Lee et al., 2018, Hsu et al., 2020.).

Thank you again for your review.

Reviewer 2 Report

Comments and Suggestions for Authors

The paper describes a deep learning with transfer learning for trying to solve a high computationally problem: sound speed reconstruction. Having two datasets, a real-life dataset and a breast cancer dataset, the algorithms are cross verified and assessed.

The methodology is sufficient described, the results are clear, and the discussion covers the methodology and the results. Conclusions are clear, and sustained by the results. 

There are minor concerns that would be nice to be addressed:

1. Since the methodology relies on transfer learning and is quite complex, a way to share the method would be nice: the algorithm itself, or the trained networks resulted, so anybody could reproduce the experiment.

1a. Maybe a more extensive way of describing which layers you have transferred from ResNet, and if any of them are fixed during the retraining.

2. Title should not be ended with period (.) lines 4,117,132, etc.

3. The sentence on line 41 misses the subject (might be techniques), or it is unclear.

4. On line 71, I would call DNNs stochastic, rather than data-driven, but it's kind of clear what the author tries to express.

5. Paragraph from lines 71-81 lacks references for many papers.

6. I would find it more practical to have the references after the "et. al" formulation rather than at the end of that sentence.

7. At line 166, the explanation "is shown below" would improve if you used the formula number.

8. The first sentence on line 346 brings no information.

Comments on the Quality of English Language

With minor editing, English is clear.

Author Response

Comment 1: (1) Since the methodology relies on transfer learning and is quite complex, a way to share the method would be nice: the algorithm itself, or the trained networks resulted, so anybody could reproduce the experiment. (1a) Maybe a more extensive way of describing which layers you have transferred from ResNet, and if any of them are fixed during the retraining.

Response 1: Thank you for pointing this out and sorry for unclear explanation. In this paper, transfer learning was not used. ResNet was used as just a basement of the network architecture, and the weight was not transferred. To clarify this point, I added an explanation that the transfer learning was not used. (Page 4, lines 183-184, highlighted.)

Comment 2: Title should not be ended with period (.) lines 4,117,132, etc.

Response 2: Thank you for pointing this out. We corrected this point.

Comment 3: The sentence on line 41 misses the subject (might be techniques), or it is unclear.

Response 3: Thank you for pointing this out. We corrected the sentence by using ‘techniques’ you suggested. (Page 1, lines 41-42, highlighted.)

Comment 4: On line 71, I would call DNNs stochastic, rather than data-driven, but it's kind of clear what the author tries to express.

Response 4: Thank you for pointing this out. We have removed the ambiguous sentence containing the word 'data-driven', as there are other aspects to DNN than data-driven. (Page 2, lines 72-73, highlighted.)

Comment 5: Paragraph from lines 71-81 lacks references for many papers.

Response 5: As suggested, we have inserted references to support our statements. The added references are (Fan et al., 2022, Prasad et al., 2022, Feigin et al., 2019, Farahani et al., 2020, Jush et al, 2023.), which align with the context and content of our discussion. (Page 2, lines 73, 77-80, highlighted.)

Comment 6: I would find it more practical to have the references after the "et. al" formulation rather than at the end of that sentence.

Response 6: Thank you for pointing this out. I corrected the format as you suggested.

Comment 7: At line 166, the explanation "is shown below" would improve if you used the formula number.

Response 7: Thank you for pointing this out. I added formula numbers. (Page 4, lines 167, 168, highlighted.)

Comment 8: The first sentence on line 346 brings no information.

Response 8: Thank you for pointing this out. This sentence was connected to the subsequent sentence to clarify the information to be shown. (Page 11, lines 347-350, highlighted.)

Thank you again for your review.